# The gap between knowledge and action in Zimbabwe: The limits of individual awareness in the face of structural violence in cholera endemicity

**Saime Ulucayli** [1]*, **Afet Arkut**[2], **Anele Lunga**[3]

**1** Faculty of Economics and Administrative Sciences, Cyprus International University, Lefkosa, TRNC, via Mersin 10, Turkey, **2** Faculty of Health Sciences, Cyprus International University, Lefkosa, TRNC, via Mersin 10, Turkey, **3** Institute of Graduate Studies and Research, Cyprus International University, Lefkosa, TRNC, via Mersin 10, Turkey

* sulucayli@ciu.edu.tr

## Abstract

### Background/Objectives

Zimbabwe faces a protracted cholera crisis exacerbated by El Niño-induced droughts, transitioning from episodic outbreaks to an endemic public health emergency. While recurrent outbreaks are often attributed to poor community awareness, this study moves beyond the conventional Knowledge, Attitudes, and Practices (KAP) paradigm. Grounded in the Integrated Behavioral Model for WASH and the theory of structural violence, we investigate the structural determinants of the knowledge-practice gap to understand how infrastructural disparities constrain cholera prevention.

### Methods

A quantitative cross-sectional study was conducted among 307 adult residents in rural and urban Zimbabwe using a structured online survey. To address the multidimensional nature of WASH behaviors, Exploratory Factor Analysis was performed to decompose the practice scale into two distinct dimensions: Individual Agency and Structural Access. Multiple linear regression analyses were applied to identify predictors for each dimension independently.

### Results

Participants demonstrated high overall knowledge of cholera prevention. However, regression analysis revealed a striking divergence: while knowledge significantly predicted individual agency behaviors, it had no statistically significant effect on structural access practices. Residence and socio-economic factors heavily dictated structural access. Additionally, women exhibited higher practice scores, reflecting

**Data availability statement:** In compliance with PLOS ONE's policy, we have provided the minimal, fully anonymized data set as a CSV file (S1_Data.csv) under the Supporting Information category. This includes all raw data used for the statistical analyses.

**Funding:** The author(s) received no specific funding for this work.

**Competing interests:** The authors have declared that no competing interests exist.

**Abbreviations:** KAP, Knowledge, Attitudes, and Practices; WASH, Water, Sanitation, Hygiene; IBM-WASH, Integrated Behavioral Model for Water, Sanitation, and Hygiene.

the disproportionate burden of sanitation insecurity rather than mere adherence to domestic gender roles.

## Conclusions

The persistent knowledge-practice gap in Zimbabwe's cholera endemic is not a behavioral failure but a manifestation of structural violence. Health education campaigns have reached saturation; individuals know how to protect themselves but are structurally paralyzed by decaying municipal services. Sustainable elimination requires shifting policy to climate-resilient WASH infrastructure.

---

## 1. Introduction

Cholera is a devastating public health problem, which is caused by *Vibrio cholerae* [1,2]. This bacterium thrives in unsanitary, crowded living conditions with poor sanitation and is primarily represented by two toxigenic serogroups, O1 and O139 [3,4]. *While Vibrio cholerae* is transmitted through the fecal-oral route- often via contaminated water or food- traditional public health narratives that focus solely on individual behavior change have limited value in environments where high-quality water and sanitation services are fundamentally absent. In such contexts, focusing on 'correct' behaviors without addressing the underlying structural deprivation can be counterproductive, as access to safe infrastructure is a fundamental human right and a state obligation that goes beyond simple biological prevention. Additionally, consuming undercooked shellfish contaminated with *Vibrio cholera*e can also spread the disease. Diagnosis is made by detecting the *Vibrio cholerae* bacterium in stool samples taken from infected individuals [2].

*Vibrio cholerae* serogroups O1 and O139, in particular, secrete toxins and cause clinical symptoms such as watery diarrhea, nausea, vomiting, abdominal cramps, and thirst [2,3]. The disease can range from mild symptoms to life-threatening dehydration [4]. One of the most common symptoms associated with cholera is watery diarrhea, known as "rice water stool." Complications of cholera include dehydration, low blood pressure, hypoglycemia, electrolyte imbalance, kidney failure, shock, coma, and death [2].

Cholera treatment relies on oral and intravenous therapy to replace water and electrolytes lost through the digestive system [2,4]. Rapid and effective treatment reduces the mortality rate from over 50% to below 1% [4].

Risk factors for the disease include poor sanitation and hygiene conditions, low stomach acid (achlorhydria), and household contact. The disease progresses more severely in immunocompromised patients or individuals without access to rehydration and medical treatment [2].

The increasing number of strains exhibiting multidrug resistance highlights the growing importance of rehydration and, in particular, preventive measures. Improving access to clean and safe drinking water, ensuring proper sanitation, and promoting

sanitation, and hygiene (WASH) practices are fundamental to preventing endemic cholera and outbreaks, especially in areas with limited access to healthcare [4].

The WHO aims to reduce cholera-related deaths by 90% by 2030 and has developed a roadmap to achieve this. Despite this, approximately among 1.3 and 4.0 million cases of cholera and between 21,000 and 143,000 deaths reported globally each year [2]. However, the global burden of cholera is not evenly distributed, with the highest number of cases and deaths occurring in Africa, Asia, and the Middle East [3].

Surveillance studies, sewage management, water sanitation, hygiene practices, awareness campaigns, rapid and effective treatment, and oral cholera vaccines are crucial in the prevention and management of cholera [3].

Traditional public health approaches have often focused on individual behavior change and lack of knowledge in preventing waterborne diseases such as cholera. However, recurring outbreaks in Zimbabwe over the past decade have painfully demonstrated that the problem is not merely one of "health literacy." As Chigudu (2020) points out in his study, cholera in Zimbabwe is a "political epidemic" created by the collapse of the state#39;s capacity to maintain public services and the neglect of urban infrastructure, rather than the spread of a biological pathogen [5]. It is not that individuals "do not know" hygiene rules, but rather, as Farmer (2006) conceptualized as "structural violence," poverty, social inequality, and institutional deprivation paralyze individuals' capacity to protect themselves [6].

This structural collapse is not unique to Zimbabwe. According to the analysis with the examining WASH (Water, Sanitation, Hygiene) practices across South Africa revealed that, the biggest barrier to hygiene behaviors in rural communities is not "unwillingness" but "geographical inequalities" and chronic deficiencies in infrastructure investments. Knowledge, Attitude, and Practice (KAP) studies in the current literature are generally successful in measuring what individuals know, however, they are unable to adequately explain why people do not act on their knowledge. Classical behavioral models are unable to explain "structural non-compliance," particularly in crisis situations where sewage problems and water outages have become common place [7].

## 2. Materials and methods

a. Study design

A quantitative cross-sectional research design was employed for this study. The chosen methodology is structured to explore the knowledge, attitudes, and practices (KAP) related to water, sanitation, and hygiene (WASH) in the prevention of cholera outbreaks in Zimbabwe.

b. Study population and sampling technique

A non-probability convenience sampling method was employed via an online survey platform. While the study was open to all adult residents of Zimbabwe, the digital nature of the data collection resulted in a sample characterized by high formal education (60.6% university/college graduates) and consistent internet access. This sample distribution provided a unique analytical advantage for testing our hypothesis: it allowed us to examine whether individuals who possess both the education and the means to access health information are still constrained by systemic factors. By focusing on this group, we were able to isolate the impact of structural violence from individual awareness, demonstrating that even high health literacy cannot overcome the barriers of a decaying municipal infrastructure. Data collection was conducted over a three-month period, beginning in November 2025 and concluding in January 2026.

c. Data tools and analysis

The questionnaire occurred from key demographic information's and household composition. The questionnaire assessed knowledge of cholera symptoms, causes, transmission routes, and prevention methods such as hand washing and use of treated water. Attitudinal components explored participants' perceptions of cholera severity, willingness to seek

treatment at Cholera Treatment Centers, readiness to notify health authorities in the event of a cholera case, and likelihood of administering Oral Re-hydration Solution at home before seeking medical care. In addition, the survey included questions on household WASH practices, such as the source and treatment of drinking water, methods of human waste disposal, types of latrines or toilets used, and whether respondents routinely washed their hands with soap after using the toilet.

d. Ethical Approval:

"This study was conducted in accordance with the Declaration of Helsinki. Ethical approval was obtained from [Cyprus International University/Ethical Committee] (Approval number: EKK25-26/06/09). Informed consent was obtained from all subjects involved in the study via an electronic consent form prior to starting the online survey."

Informed Consent: All participants provided informed consent before participating in the study. Because the study was conducted via an online platform, electronic written consent was obtained. Participants were required to read a detailed information sheet on the first page of the Google Form.

## 3. Results

Table 1 shows the sociodemographic characteristics of the participants. The results of the study revealed that females comprised the majority of the participants with 56,7% (n = 174), whereas males population followed with 36, 2% (n = 111). The highest number of participants were from the ages between 26–35, with 59.%. Most of the participants are graduated from college/university with 60.6% (n = 186). Also, most of the participants, 58% (n = 178), lived with a nuclear family; 22.5% (n = 69) lived alone; and 19.5% (n = 60) resided with extended family. Additionally, number of participants were lived in urban areas, with 53.7% (n = 165), 46,3% (n = 142) lived in in rural areas.

"Note: The 'Other' category in the gender section includes participants who preferred not to self-identify or chose not to disclose their gender".

**Table 1. Descriptive Statistics of Participants (n = 307).**

|  | Variable | Frequency (n) | Percent (%) |
|---|---|---|---|
| **Gender** | Female | 174 | 56.7 |
|  | Male | 111 | 36.2 |
|  | Other | 22 | 7.2 |
| **Age** | 18–25 | 47 | 15.3 |
|  | 26–35 | 181 | 59.0 |
|  | 36–45 | 53 | 17.3 |
|  | 46–55 | 19 | 6.2 |
|  | 56+ | 7 | 2.3 |
| **Education** | College/University | 186 | 60.6 |
|  | No formal | 23 | 7.5 |
|  | Other | 20 | 6.5 |
|  | Primary | 12 | 3.9 |
|  | Secondary | 66 | 21.5 |
| **Residence** | Rural | 142 | 46.3 |
|  | Urban | 165 | 53.7 |
| **Living status** | Nuclear family | 178 | 58 |
|  | Extended family | 60 | 19,5 |
|  | Alone | 69 | 22,5 |

Table 2 represents the descriptive and reliability statistics for the Knowledge, Attitude and Practice (KAP) scales used in this study. The majority of respondents showed high knowledge of WASH and cholera prevention, as evidenced by the Knowledge scale#39;s relatively high mean score (M = 1.89 ± 0.15) and strong negative skewness. The Attitude scale#39;s moderate mean value (M = 2.34 ± 0.89) and small positive skewness indicate that respondents' attitudes toward WASH activities vary substantially. The highest mean score (M = 3.24 ± 0.70) on the Practice scale indicated generally positive WASH practices. The Knowledge (α = .782) and Attitude (α = .878) measures had adequate internal consistency according to Cronbach#39;s alpha values, however the Practice scale had poorer reliability (α = .599), which should be taken into account when interpreting practice-related results.

The items were clustered under two main factors 'Individual Agency' factor, representing individual hygiene behaviors and explaining 36.13% of the variance (PRACT1, 2, 3, 5). 'Structural Access' factor, representing household infrastructural facilities and explaining 13.9% of the variance (PRACT8, 9). This differentiation statistically validates that individual motivation and infrastructural constraints are two independent dimensions in the cholera intervention in Zimbabwe, as summarized in Table 3."

"To test the structural validity of the Practice scale, Principal Component Analysis (PCA) was performed using the Varimax rotation method. The analysis revealed a sample adequacy KMO value of 0.734 and a significant Bartlett sphericity test (p < .001), indicating that the data are suitable for factor analysis. The results regarding the suitability of the data for this analysis are presented in Table 4."

To determine the sociodemographic predictors of the participants' knowledge, attitude, and practice levels, multiple linear regression analyses were performed, and the results are detailed in **Table 5**. Despite explaining a small percentage of the variance, the model predicting knowledge was statistically significant ($R^2 = .092$, F (5, 301) = 6.133, p < .001). Education was the only significant predictor (β = −.264, t = −4.484, p < .001), suggesting that lower knowledge scores were linked

**Table 2. Descriptive and Reliability Statistics of KAP Scales.**

| Scale | Mean | SD | Skewness | Cronbach α |
|---|---|---|---|---|
| Knowledge | 1.8897 | 0.15098 | −1.933 | .782 |
| Attitude | 2.3393 | 0.89256 | .548 | .878 |
| Practice | 3.2432 | 0.69827 | −.595 | .599 |

**Table 3. Results of Exploratory Factor Analysis for the Practice Scale.**

| Factors and Items | Factor Loading | Eigenvalue Rotated | Explained Variance (%) | Cumulative Variance (%) |
|---|---|---|---|---|
| **Factor 1: Individual Agency** | | 3.25 | 36.13 | 36.13 |
| PRACT1 | 0.782 | | | |
| PRACT2 | 0.714 | | | |
| PRACT3 | 0.816 | | | |
| PRACT5 | 0.777 | | | |
| **Factor 2: Structural Access** | | 1.25 | 13.92 | 50.05 |
| PRACT8 | 0.717 | | | |
| PRACT9 | 0.571 | | | |
| **KMO Measure of Sampling Adequacy** | **0.734** | | | |
| Bartlett's Test of Sphericity | **p < .001** | | | |

**Table 4. Test Results Regarding the Suitability of Data for Factor Analysis.**

| Statistical Tests | Findings |
|---|---|
| Kaiser-Meyer-Olkin (KMO) Measure of Sampling Adequacy | 0.734 |
| Bartlett's Test of Sphericity Approx. Chi-Square ($x^2$) | [808.707] |
| Bartlett's Test of Sphericity Degrees of Freedom (df) | [36] |
| Bartlett's Test of Sphericity Significance (p) | < 0.001 |

**Table 5. Multiple Linear Regression Results for Knowledge, Attitude, and Practice.**

Coefficients[a]

| Model | | Unstandardized Coefficients | | Standardized Coefficients | t | Sig. | Collinearity Statistics | |
|---|---|---|---|---|---|---|---|---|
| | | B | Std. Error | Beta | | | Tolerance | VIF |
| 1 | (Constant) | 1.184 | .828 | | 1.431 | .154 | | |
| | Gender | −.103 | .092 | −.065 | −1.120 | .264 | .956 | 1.046 |
| | Age | −.041 | .069 | −.035 | −.592 | .554 | .916 | 1.092 |
| | Education | −.029 | .038 | −.049 | −.772 | .441 | .813 | 1.230 |
| | Whoareyoulivingwith | −.152 | .072 | −.126 | −2.131 | .034 | .924 | 1.082 |
| | Residence | −.104 | .125 | −.052 | −.833 | .405 | .829 | 1.206 |
| | KnowTotal | −.013 | .026 | −.029 | −.487 | .627 | .908 | 1.102 |

a. Dependent Variable: REGR factor score 2 for analysis 1.

to higher educational achievement. Gender, age, living situation, and residence were among the other variables that did not exhibit any significant effects (all p>.05).

Similar statistical significance was attained by the regression model for attitude ($R^2$=.101, F (5, 301) = 6.779, p<.001). Furthermore, living status showed a slight but significant negative correlation (β=−.126, t=−2.229, p=.027), while education continued to be a substantial negative predictor (β=−.215, t=−3.672, p<.001). Attitude scores were not significantly impacted by residence, age, or gender.

In terms of the three models, the one that predicted Practice had the highest explanatory power ($R^2$=.320, F (5, 301) = 28.298, p<.001). Residence was shown to be the most significant predictor in this model (β=−.516, t=−9.924, p<.001), suggesting a significant detrimental impact on practice levels. Education also had a slight but significant negative impact (β=−.101, t=−1.979, p=.049). The remaining predictors (all p>.05) were not statistically significant.

Gender and age were consistently non-significant predictors in all models, indicating that they had no bearing on the sample#39;s differences in knowledge, attitude, or practice. The range of Variance Inflation Factor (VIF) values was 1.043 to 1.198, indicating that multicollinearity issues were not present.

## 4. Discussion

The findings revealed a positive and significant relationship between knowledge level and attitude (r=.189, p=.001). However, upon decomposing the 'practice' variable into behavioral and structural dimensions—guided by the Integrated Behavioral Model for Water, Sanitation, and Hygiene (IBM-WASH) —a striking divergence emerged (Table 3,4). While knowledge significantly predicted individual hygiene behaviors (p<.001), our refined regression model conclusively demonstrated that total knowledge had no statistically significant effect on structural WASH practices (p=.627) (Table 5). Furthermore, our analysis showed that the 'residence' variable remained a critical predictor, but notably, 'living

arrangement' also emerged as a significant factor (p = .034), suggesting that household density and composition further constrain access to limited infrastructure. This indicates that the fight against cholera in Zimbabwe is shaped more by infrastructural access, household constraints, and environmental factors than by individual awareness.

The predominance of highly educated participants in our sample (60.6% college/university graduates) strengthens the validity of our 'knowledge-practice gap' findings. It confirms that even among those with the highest capacity for health literacy and information access, structural violence remains the primary barrier to cholera prevention. Our study#39;s most notable result is that high knowledge levels fail to translate into adequate structural WASH practices. This situation, referred to in the literature as the 'knowledge-practice gap,' is consistent with other studies conducted in rural areas of Zimbabwe and Southern Africa [8]. The statistical insignificance of knowledge in predicting structural access (p = .627) reinforces the argument that awareness alone cannot overcome physical barriers. This is consistent with Oberg's (2019) finding that behavior change messaging often fails by focusing on educating individuals about biological dangers while ignoring the fact that they lack the basic services required to act on that knowledge. Such a 'behavior change' narrative has little value in environments where water and sanitation services are non-existent or of poor quality [9]. For example, previous studies reported that mothers knew the importance of handwashing ('doers'), but economic difficulties in accessing soap and the lack of handwashing stations prevented this knowledge from translating into behavior [10]. Similarly, a study conducted on adolescents in Haiti reported that despite high levels of knowledge about cholera, hygiene practices were inadequate due to physical difficulties in accessing clean water and a lack of soap. This disconnect, also observed in our study, suggests that although individuals know what to do, they are unable to translate this knowledge into action due to water scarcity caused by climate change and infrastructure deficiencies, as also highlighted in research in Zimbabwe [ 11,12]. In contrast, studies conducted in Lebanon found that higher levels of knowledge were associated with better practices. This difference may stem from socio-economic conditions or access to water in the Lebanese sample allowing for greater individual choice compared to rural Zimbabwe [3,13].

However, framing this disconnect merely as a behavioral gap is insufficient. Instead, as posited by Farmer et al. (2006), this phenomenon is a clear manifestation of 'structural violence' [6]. Cholera is not simply a microbe problem; it is a form of systemic violence against the poor. The inability of knowledgeable individuals to adopt safe WASH practices—statistically proven by our non-significant p-value for knowledge—is a consequence of social, political, and economic arrangements that systematically put these vulnerable populations in harm's way.

The overwhelming influence of the 'residence' variable underscores how geographical inequalities dictate health outcomes. As highlighted by Chigudu (2020), the recurrent cholera outbreaks in Zimbabwe are deeply rooted in a 'political epidemiology.' The crisis is a direct result of the state withdrawing municipal services and the historical decay of public health infrastructure, particularly in Harare and its high-density suburbs. Individuals living in these areas are structurally excluded from reliable municipal services, rendering their high health literacy largely ineffective against the infrastructural barriers they confront daily [5].

Our findings also revealed that females demonstrated higher knowledge and practice scores. While traditionally attributed to socially constructed "gender roles," this finding is more accurately explained through the lens of 'sanitation insecurity.' Women bear the disproportionate burden of navigating water scarcity and inadequate sanitation, experiencing heightened physical exertion, insecurity, and psychosocial stress when accessing water. Their higher scores, therefore, reflect a forced adaptation to systemic infrastructural failures and the burden of insecurity rather than a simple adherence to domestic responsibilities.

This structural vulnerability has been critically exacerbated by recent climate shocks. The 2023–2025 epidemiological landscape in Zimbabwe has been severely impacted by an El Niño-induced drought, which depleted safe water sources and forced communities to rely on contaminated alternatives [14]. According to UNICEF reports, the protracted 2023–2024 outbreak resulted in over 34,550 cases and 719 deaths, followed by a resurgence in late 2024 and early 2025 driven by these climatic extremes [15]. Furthermore, UNICEF and recent epidemiological studies highlight that due to the

persistent deterioration of WASH infrastructure, cholera has now become endemic in the country [16]. While Oral Cholera Vaccines (OCV) are often deployed as a reactive measure in Zimbabwe, this study focuses on the fundamental structural drivers—namely water and sanitation infrastructure—which provide a more sustainable solution than transient vaccination campaigns. As argued by Hichilema and Ghebreyesus (2025), although tools like vaccines are available, the failure to eliminate cholera is fundamentally a political and infrastructural issue rather than a lack of medical interventions. Therefore, public health strategies must prioritize climate-resilient infrastructure to dismantle the structural violence sustaining this endemic [17,18]. Mintah et al. (2024) note that extreme droughts and political constraints in water supply are primary drivers of this endemicity. This localized crisis mirrors a broader regional threat, as Bekele et al. (2025) emphasize that El Niño-driven climate changes are fundamentally altering the geography of cholera across Africa, making it a persistent climate change crisis rather than an episodic emergency.

In contrast, studies conducted in regions like Lebanon found that higher levels of knowledge were associated with better practices. This difference clearly stems from variations in socio-economic conditions and structural access. Ultimately, this disconnect, also observed in our study, suggests that although individuals know what to do, they are structurally paralyzed. Public health interventions must therefore move beyond individual awareness campaigns and prioritize climate-resilient, equity-driven WASH infrastructure investments to dismantle the structural violence sustaining this endemic.

## 5. Conclusions

Findings of our study demonstrate that while health education effectively enhances individual hygiene agency, it fundamentally fails to bridge the knowledge-practice gap when individuals confront severe infrastructural deficits. The inability to adopt safe WASH practices is not a reflection of community ignorance, but rather a direct consequence of geographical inequalities, sanitation insecurity, and a decaying municipal infrastructure that disproportionately burdens women and marginalized populations. Amplified by the severe 2023–2025 El Niño-induced droughts and subsequent water shortages, these systemic failures have entrenched cholera as an endemic public health crisis rather than a sporadic outbreak.

Ultimately, awareness alone cannot substitute for a functional water supply. The statistical divergence between individual actions and structural realities in our data proves that educational campaigns have reached their saturation point. Future policies, local governance, and global health interventions must move beyond redundant behavioral nudges and focus on dismantling the structural barriers that paralyze knowledgeable individuals. Prioritizing climate-resilient, gender-sensitive, and equitable WASH infrastructure investments is the only sustainable pathway to ending cholera transmission and restoring health equity in Zimbabwe.

## Supporting information

**S1 Table. Survey Questionnaire.**
(DOCX)

## Acknowledgments

This manuscript/study, based on "The Gap Between Knowledge and Action in Zimbabwe: The Limits of Individual Awareness in the Face of Structural Violence in Cholera Endemicity." The authors have reviewed and edited the output and take full responsibility for the content of this publication.

## Author contributions

**Methodology:** Saime Ulucayli, AFET ARKUT.
**Writing – original draft:** Saime Ulucayli, AFET ARKUT, ANELE LUNGA.
**Writing – review & editing:** Saime Ulucayli, AFET ARKUT, ANELE LUNGA.

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
