## [Decision Letter · Decision Letter 0]

30 Mar 2026

Dear Dr. Ulucayli,

As the corresponding author, your ORCID iD is verified in the submission system and will appear in the published article. PLOS supports the use of ORCID, and we encourage all coauthors to register for an ORCID iD and use it as well. Please encourage your coauthors to verify their ORCID iD within the submission system before final acceptance, as unverified ORCID iDs will not appear in the published article. *Only* the individual author can complete the verification step; PLOS staff the individual author can complete the verification step; PLOS staff the individual author can complete the verification step; PLOS staff the individual author can complete the verification step; PLOS staff *cannot* verify ORCID iDs on behalf of authors.verify ORCID iDs on behalf of authors.verify ORCID iDs on behalf of authors.verify ORCID iDs on behalf of authors.

We look forward to receiving your revised manuscript.

Kind regards,

Alison Parker

Academic Editor

PLOS One

2. During your revisions, please confirm whether the wording in the title is correct and update it in the manuscript file and online submission information if needed. Specifically, Please change your title from "To the Editorial Board, PLOS ONE The Gap Between Knowledge and Action in Zimbabwe: The Limits of Individual Awareness in the Face of Structural Violence in Cholera Endemicity"" to "The Gap Between Knowledge and Action in Zimbabwe: The Limits of Individual Awareness in the Face of Structural Violence in Cholera Endemicity.

Reviewers' comments:

Reviewer's Responses to Questions

**Comments to the Author**

1. Is the manuscript technically sound, and do the data support the conclusions?

Reviewer #1: Yes

Reviewer #2: Yes

2. Has the statistical analysis been performed appropriately and rigorously?

Reviewer #1: I Don't Know

Reviewer #2: I Don't Know

3. Have the authors made all data underlying the findings in their manuscript fully available?

Reviewer #1: Yes

Reviewer #2: Yes

4. Is the manuscript presented in an intelligible fashion and written in standard English?

Reviewer #1: Yes

Reviewer #2: Yes

Reviewer #1: Comments:

This is a well written article that clearly describes the current cholera situation on the ground. The manuscript effectively highlights that the ongoing outbreak is largely driven by the lack of adequate WASH (Water, Sanitation, and Hygiene) facilities. The contextual explanations are strong and help the reader understand the underlying structural challenges contributing to transmission.

Minor Edits:

1. The correct scientific name is Vibrio cholerae, not Vibrio cholera. Please update this throughout the manuscript.

Reviewer #2: General comments

The message of this article is important and unfortunately needs to be repeated often. You could reference qualitative research that has covered the same ground, for example Oberg (2019) who found that behaviour change messaging often focuses on educating people on “the biological dangers of shit, assuming that lack of scientific knowledge is the problem”. It’s useful to have the proof of this messaging through the survey data collected here.

I believe you can make the point even more strongly that the ‘behaviour change’ narrative has little value in an environment of no or poor-quality water and sanitation services. This could include referring to the obvious fact that water and sanitation services do not only make it possible for households to engage in better hygiene behaviour to tackle cholera, but also many other health and other risks. (No need to say all this, but also: better services contribute to meeting government obligations to realise the human rights to water and sanitation, relieve particularly women and (girl) children of onerous tasks, increase opportunities for income generation, improve and protect the environment etc., including supporting governments’ obligations to realise the human right to a safe environment.) In short - this behaviour change narrative is not only damaging for efforts to address the risk of cholera.

The article needs a grammar edit – there are a couple of typos (for example first sentence of the introduction needs attention).

I am not sufficiently well-versed in statistical analysis to respond to that aspect of the paper, but I have the following questions:

Line 89 Study population

I don’t understand the rationale, or perhaps I don’t understand the selection criteria. The way it is currently worded, you have selected people over 18 with high-level education and internet access, therefore living without adequate water, sanitation and hygiene. But I think what you mean is that they both have high level of education AND water, sanitation and hygiene challenges? Can you reword this to be clearer. Likewise internet access –is the only relevance to the study population that the respondents are able to fill in the survey online? Perhaps clarify whether this has any other relevance.

Also on the study population – perhaps you could also clarify why focused more on the population with higher education. How would the study be different if you looked at people without higher education? Would you expect to see different results? Is there a population group that does benefit from behaviour change messaging over better service provision?

Line 102 – I am interested that cholera vaccinations in this list of treatments is not referred to at all, despite this being a WHO preference over sanitation interventions, see (Hichilema and Ghebreyesus, 2025). Are they not prevalent in Zimbabwe? Would that not be something that people had taken and would influence cholera infections? (Let me stress that I personally do not think that cholera vaccinations are a better solution than water and sanitation, just that they are touted as a solution, so it is interesting that this is not mentioned in the survey).

Table 1 – Under Gender (Male, Female, Other) Did 7.2% refer to themselves as ‘other’ or just not respond to this question? If the former, what constitutes ‘other’ in this context?

Also – wealth was mentioned as a factor for choosing the study population, but that is not mentioned here. Do you have relevant wealth data? If not, perhaps take it out of the definition of the study population.

Hichilema, H. & Ghebreyesus, T. A. 2025. Cholera is spreading fast, yet it can be stopped. Why haven’t we consigned it to history? The Guardian, 25 October 2025.

Oberg, A. 2019. Problematizing Urban Shit(ting): Representing Human Waste as a Problem. International Journal of Urban and Regional Research, 43, 377-392.

.

Reviewer #1: No

Reviewer #2: No

---

## [Author Response · Author response to Decision Letter 1]

6 Apr 2026

Response to Reviewers

Manuscript ID: PONE-D-26-12022

Title: The Gap Between Knowledge and Action in Zimbabwe: The Limits of Individual Awareness in the Face of Structural Violence in Cholera Endemicity

Authors: Saime Ulucayli, Afet Arkut, Anele Lunga

Dear Dr. Parker and the Editorial Board of PLOS ONE,

We would like to thank the editorial team and the reviewers for their insightful comments and constructive feedback. We have addressed each point raised and believe the revisions have significantly strengthened the manuscript. All changes are highlighted using the Track Changes feature in the revised manuscript.

Below is our point-by-point response:

Part 1: Academic Editor’s Requirements

1. Title and Header Correction:

• Comment: Change your title from "To the Editorial Board, PLOS ONE..." to the actual title.

• Response: We have removed the editorial salutation from the header. The manuscript now begins directly with the formal title.

(See Lines 1-3)

2. Data Availability Statement:

• Comment: Address restrictions to data sharing.

• Response: In compliance with PLOS ONE’s policy, we have provided the minimal, fully anonymized data set as a CSV file (S1_Data.csv) under the Supporting Information category. This includes all raw data used for the statistical analyses.

(See Data Availability Statement, Lines 394-396)

3. Ethics Statement Placement:

• Comment: Ethics statement should only appear in the Methods section.

• Response: The ethics statement has been removed from the end of the manuscript and is now exclusively located within the Methods section under the "Ethical Approval" subsection.

(See Lines 127-134)

Part 2: Response to Reviewer #1

1. Scientific Nomenclature:

• Comment: Use the correct name Vibrio cholerae.

• Response: We have corrected "Vibrio cholera" to Vibrio cholerae (italicized) throughout the entire manuscript.

(See Lines 33, 35, 41, 42 and 44)

Part 3: Response to Reviewer #2

1. Grammar and Opening Sentence:

• Comment: The first sentence of the introduction needs attention.

• Response: We have rewritten the first sentence to correct the grammar and to better introduce the biological and structural context of the study.

(See Lines 33-43)

2. 'Behaviour Change' Narrative and Human Rights:

• Comment: Strengthen the argument that the ‘behaviour change’ narrative has little value in infrastructure-deprived environments.

• Response: We have revised the Introduction to emphasize that WASH infrastructure is a fundamental human right. We explicitly state that focusing on individual behavior without addressing structural deprivation can be counterproductive.

(See Lines 33-43)

3. Study Population Rationale (Education and Internet):

• Comment: Clarify why a sample with high education and internet access was examined.

• Response: We have added a clarification in the Methods section. We argue that this sample provides an "analytical advantage": it allowed us to examine whether individuals who possess both the education and the means to access health information are still constrained by systemic factors. By focusing on this group, we were able to isolate the impact of structural violence from individual awareness, demonstrating that even high health literacy cannot overcome the barriers of a decaying municipal infrastructure.

(See Lines 106-116)

4. Oral Cholera Vaccines (OCV):

• Comment: Address why vaccinations are not referred to.

• Response: We have integrated a discussion on OCV, citing Hichilema and Ghebreyesus (2025). We clarify that while vaccines are vital reactive tools, our focus is on the more sustainable solution of long-term infrastructural investment.

(See Lines 299-305)

5. Qualitative Literature (Oberg, 2019):

• Comment: Refer to qualitative research like Oberg (2019).

• Response: We have incorporated Oberg (2019) into the Discussion to support our findings regarding the failure of individual-focused behavior change messaging in the absence of basic services.

(See Lines 255-258)

6. Table 1 "Other" Category:

• Comment: Clarify the "Other" gender category.

• Response: We have added a clarifying note below Table 1, specifying that this category includes participants who chose not to self-identify or preferred not to disclose their gender.

(See Lines 183-185)

We hope these revisions meet the requirements for publication in PLOS ONE.

Sincerely,

Dr. Saime Ulucayli (on behalf of all authors)

Corresponding Author Cyprus International University

---

## [Decision Letter · Decision Letter 1]

8 Apr 2026

The Gap Between Knowledge and Action in Zimbabwe: The Limits of Individual Awareness in the Face of Structural Violence in Cholera Endemicity

PONE-D-26-12022R1

Dear Dr. Ulucayli,

We’re pleased to inform you that your manuscript has been judged scientifically suitable for publication and will be formally accepted for publication once it meets all outstanding technical requirements.

Kind regards,

Alison Parker

Academic Editor

PLOS One

Additional Editor Comments (optional):

Reviewers' comments:

Reviewer's Responses to Questions

**Comments to the Author**

Reviewer #2: All comments have been addressed

2. Is the manuscript technically sound, and do the data support the conclusions?

Reviewer #2: (No Response)

3. Has the statistical analysis been performed appropriately and rigorously?

Reviewer #2: (No Response)

4. Have the authors made all data underlying the findings in their manuscript fully available?

Reviewer #2: (No Response)

5. Is the manuscript presented in an intelligible fashion and written in standard English?

Reviewer #2: (No Response)

Reviewer #2: (No Response)

.

Reviewer #2: No

---

## [Editor Report · Acceptance letter]

PONE-D-26-12022R1

PLOS One

Dear Dr. Ulucayli,

I'm pleased to inform you that your manuscript has been deemed suitable for publication in PLOS One. Congratulations! Your manuscript is now being handed over to our production team.

Kind regards,

on behalf of

Dr. Alison Parker

Academic Editor

PLOS One